# Effectiveness of the Influence of Selected Essential Oils on the Growth of Parasitic *Fusarium* Isolated from Wheat Kernels from Central Europe

**DOI:** 10.3390/molecules26216488

**Published:** 2021-10-27

**Authors:** Teresa Krzyśko-Łupicka, Sławomir Sokół, Monika Sporek, Anna Piekarska-Stachowiak, Weronika Walkowiak-Lubczyk, Adam Sudoł

**Affiliations:** 1Institute of Environmental Engineering and Biotechnology, Faculty of Natural and Technical Sciences, University of Opole, Kominka 6/6A, 45-035 Opole, Poland; 2Institute of Biology, Faculty of Natural and Technical Sciences, University of Opole, Oleska 22, 45-052 Opole, Poland; sokol@uni.opole.pl (S.S.); mebis@uni.opole.pl (M.S.); 3Institute of Biology, Biotechnology and Environmental Protection, Faculty of Natural Sciences, University of Silesia in Katowice, Jagiellońska 28, 40-032 Katowice, Poland; anna.piekarska@us.edu.pl; 4Department of Measurements, EMITOR s.c., Olimpijska 6, 45-681 Opole, Poland; weronika.walkowiak@gmail.com

**Keywords:** *Fusarium* isolates from the German and Polish population, essential oils, the mycelial growth rate index

## Abstract

The aim of the study was to determine the effectiveness of selected seven commercial essential oils (EsO) (grapefruit, lemongrass, tea tree (TTO), thyme, verbena, cajeput, and *Litsea cubeba*) on isolates of common Central European parasitic fungal species of *Fusarium* obtained from infected wheat kernels, and to evaluate the oils as potential natural fungicides. The study was conducted in 2 stages. At each stage, the fungicidal activity of EsO (with concentrations of 0.025; 0.05; 0.125; 0.25; 0.50; 1.0, and 2.0%) against *Fusarium* spp. was evaluated using the disc plate method and zones of growth inhibition were measured. At the first stage, the fungistatic activity of EsO was evaluated against four species of *Fusarium* from the Polish population (*F. avenaceum* FAPL, *F. culmorum* FCPL, *F. graminearum* FGPL and *F. oxysporum* FOPL). The correlation coefficient between the mycelial growth rate index (T) and the fungistatic activity (FA) was calculated. At the second stage, on the basis of the mycelium growth rate index, the effectiveness of the EsO in limiting the development of *Fusarium* isolates from the German population (*F. culmorum* FC1D, *F. culmorum* FC2D, *F. graminearum* FG1D, *F. graminearum* FG2D and *F. poae* FP0D) was assessed. The first and second stage results presented as a growth rate index were then used to indicate essential oils (as potential natural fungicides) effectively limiting the development of various common Central European parasitic species *Fusarium* spp. Finally, the sensitivity of four *Fusarium* isolates from the Polish population and five *Fusarium* isolates from the German population was compared. The data were compiled in STATISTICA 13.0 (StatSoft, Inc, CA, USA) at the significance level of 0.05. *Fus**arium* isolates from the German population were generally more sensitive than those from the Polish population. The sensitivity of individual *Fusarium* species varied. Their vulnerability, regardless of the isolate origin, in order from the most to the least sensitive, is as follows: *F. culmorum*, *F. graminearum*, *F. poae*, *F. avenaceum* and *F. oxysporum*. The strongest fungicidal activity, similar to Funaben T, showed thyme oil (regardless of the concentration). Performance of citral oils (lemongrass and *Litsea cubeba*) was similar but at a concentration above 0.025%.

## 1. Introduction

According to the latest systematics (MycoBank—http://www.mycobank.org (accessed date: 10 June 2021), *Fusarium* fungi belong to the following taxa: domain *Eucaryota*, kingdom *Fungi*, phylum *Ascomycota*, subclass *Pezizomycotina*, class *Sordariomycetes*, subclass *Hypocreomycetidae*, order *Hypocreales*, and family *Nectriaceae*. *Fusarium* includes species with anamorphic development (imperfect fungi).

Species of *Fusarium* fungi are among the most diverse and widespread saprotrophs and pathogenic species in the environment. Due to their ability to produce various metabolites, mainly mycotoxins, they not only pose a serious threat to humans and animals, but also adversely affect soil fertility, biological productivity of agroecosystems, grassland, and forest ecosystems, and reduce the value of agricultural crops. They occur in many ecological niches, including cereal growing environments [1,2,3]. The *Fusarium* ear blight caused by them is considered increasingly important in many parts of Europe, including Germany, Poland, France, Denmark, Italy, and Hungary. These toxigenic polyphagous pathogens occur in varying degrees on plants every growing season in all climate zones [4] as they spread easily and attack plants at all stages of development. In central Europe, the most dominant *Fusarium* ear blight-causing species are *F. graminearum*, *F. poae, F. avenaceum, F. culmorum, F. langsethiae*, and *F. cerealis* [5,6,7].

The attempt to reduce losses caused by these phytopathogens and the increasingly evident drawbacks of synthetic fungicides [8,9] encourage a constant search for natural fungicidal substances. This is especially since the spreading phytopathogens are more likely to acquire resistance to the fungicides [10].

In recent years, many scientific centres have focused on the study of the fungicidal activity of natural plant components. Plant extracts, especially essential oils (EsO), significantly reduce the growth of fungi and can be used to eradicate *Fusarium* [1,11,12,13,14,15,16,17,18]. Currently, a research stream assuming the selection of high activity essential oils (EsO) at low concentrations and a broad spectrum of action on phytopathogens is increasingly common. The properties of EsO are particularly beneficial due to the minimal risk of pathogen resistance, relatively low toxicity to humans and the environment [19,20], as well as biodegradability and lack of bioaccumulation in the environment [21].

The chemical composition of EsO, the type and concentration of active substances and the synergistic relations between them determine the fungicidal or fungistatic mode of action [1,13,18,22]. Therefore, only some EsO are able to completely inhibit *Fusarium* growth and therefore are fungicidal. Given the varying sensitivity of fungi, it is very difficult to choose the right oil at the right concentration, especially if the essential oil is expected to effectively limit the growth of different species within *Fusarium*. Our previous study [1], whose aim was to determine the fungistatic activity of oils in relation to phytopathogens of *Fusarium* from the Polish population (*F. avenaceum* FAPL, *F. culmorum* FCPL, *F. graminearum* FGPL and *F. oxysporum* FOPL) showed that the activity of oils with a high content of thymol (thyme) and citral (*Litsea cubeba*, lemongrass and verbena) is most similar to the activity of Funaben T.

Natural populations harbour a stunning diversity of phenotypic variation for morphology, physiology, behaviour and disease susceptibility [23], and consequently sensitivity to EsO.

The aim of the study was to determine the effectiveness of selected seven commercial essential oils on isolates of common Central European parasitic fungal species of *Fusarium* obtained from infected wheat kernels and to evaluate the oils as potential natural fungicides.

## 2. Results

### 2.1. Variation in Sensitivity of Fusarium Isolates to Essential Oils

The coefficients of fungistatic activity (FA) and mycelial growth rate index (T) were found to be highly correlated (correlation coefficient FA/T was 0.99), which means that an increase in mycelial growth rate index causes a decrease in fungistatic activity. Formula (1) describes 99.15% of the variation in fungistatic activity. Therefore, the T index was used to describe the relationship between the sensitivity of the isolates and the fungicidal activity of the EsO.

The nine isolates (representing five *Fusarium* species) showed differential sensitivity to seven commercial EsO; the fungicidal and fungistatic effect is shown on the example of *F. culmorum* FC2D and *F. poae* FP0D (Figure 1 and Figure 2).

The Kruskal–Wallis test showed that the T index differences obtained for individual isolates were statistically significant. In terms of the mean T index, the individual isolates in the presence of individual EsO can be described using numbers shown in Table 1.

Cluster analysis for T index values showed that they can be divided into two groups due to their sensitivity to EsO. The highest sensitivity was observed in five isolates, including four isolates from the German population (*F. culmorum* FC1D, *F. culmorum* FC2D, *F. graminearum* FG1D, *F. graminearum* FG2D), and one isolate from the Polish population (*F. culmorum* FCPL) (Figure 3).

In the presence of grapefruit, cajeput, and tea tree oils, *F. culmorum* FC2D had a high T index mean, ranging from 44.57 to 16.17 (control 49.18) (Table 2, Figure 4a). The Appendix A contains a Table with the minimum growth rate index of the analysed oil at a minimum concentration of each considered isolates of *Fusarium*.

This indicates the low sensitivity of the isolate to the EsO. In the presence of thyme, lemongrass, *Litsea cubeba*, and verbena oils, the T index of this isolate was significantly lower, ranging from 0.00 to 3.62, which shows that the oils are highly effective. The post hoc analysis showed that no significant T index differences occurred in the presence of thyme, lemongrass, *Litsea cubeba*, and verbena oils as compared to the results obtained in the presence of Funaben T.

EsO sensitivity of *F. graminearum* FG2D (Table 2, Figure 4b) was similar to the previous isolate as the ranges of T index mean values were similar. Post hoc analysis showed that the T index in the presence of all EsO, except grapefruit, was similar to the results obtained in the presence of Funaben T, which makes this isolate essentially different from FC2D.

EsO sensitivity of *F. graminearum* FG1D (Table 2, Figure 4c) was similar to the previous isolates: FC2D and FG2D. Post hoc analysis revealed that the results are identical to FG2D.

The response of *F. culmorum* FC1D to three oils with poor activity (grapefruit, cajeput, and tea tree) was similar to the previously discussed isolates (43.10 to 13.31) (control 50.02) (Table 2, Figure 4d). Very low T index mean values in the presence of the remaining four oils (0.00 to 0.98) reveal complete inhibition of mycelial growth (0.00) in the presence of thyme, *Litsea cubeba*, and lemongrass oils, similarly to Funaben T. The activity of verbena oil was slightly weaker (0.98). This isolate showed the greatest sensitivity to EsO out of the isolates discussed so far. Post hoc analysis showed that, in the presence of all oils, except grapefruit oil, the T index did not differ from the results obtained in the presence of Funaben T for this isolate.

The response of *F. culmorum* FCPL to three oils with poor activity (grapefruit, cajeput and tea tree) was similar to the previously discussed isolates (38.19 to 11.32) (control 49.83) (Table 2, Figure 4e). Low T index values against thyme, lemongrass, *Litsea cubeba*, and verbena oil (0.00 to 1.79) prove their high effectiveness. The sensitivity of this isolate was similar to *F. culmorum* FC1D. Post hoc analysis for this isolate showed that only the activity of grapefruit and cajeput oils was significantly different from Funaben T.

The cluster analysis showed that the four remaining isolates, including one isolate from the German population (*Fusarium* poae FP0D), and three multi-species isolates from the Polish population (*F. avenaceum* FAPL, *F. graminearum* FGPL and *F. oxysporum* FOPL) had higher resistance to EsO (Figure 3).

In the presence of grapefruit, cajeput, and tea tree oils, *F. poae* FP0D had a high T index, ranging from 40.69 to 16.79 (control 49.93) (Table 2, Figure 4f). In the presence of the other four EsO (thyme, lemongrass, *Litsea cubeba*, and verbena), lower T index values were observed (0.00 to 6.02). However, these values were higher as compared to the results of the five previously discussed (more sensitive) isolates. Post hoc analysis showed that only the activity of grapefruit oil was significantly different from Funaben T.

EsO sensitivity of *F. graminearum* FGPL (Table 2, Figure 4g) was similar to *F. poae* FP0D in the presence of individual oils (the range of T index values was similar). Post hoc analysis showed that only the activity of grapefruit and cajeput oils was significantly different from Funaben T.

EsO sensitivity of *F. avenaceum* FAPL (Table 2, Figure 4h) was similar to the previous isolates (FP0D and FGPL) in the presence of individual oils (the range of T index mean values was similar). The post hoc analysis showed that no significant T index differences occurred in the presence of thyme, lemongrass, *Litsea cubeba*, verbena oils, and Funaben T.

In the presence of grapefruit, cajeput, and tea tree oils, *F. oxysporum* FOPL had a high T index, ranging from 43.52 to 13.90 (control 46.39) (Table 2, Figure 4i). In the presence of the other EsO (thyme, lemongrass, *Litsea cubeba*, and verbena), lower T index mean values were observed (1.77 to 5.09). This indicates the relatively high effectiveness. The post hoc analysis showed that no significant T index differences occurred in the presence of thyme, lemongrass, *Litsea cubeba*, verbena oils, and Funaben T. However, it was the only isolate that grew when treated with the lowest concentration of thyme oil.

The Kruskal–Wallis test (H (8, 1877) = 1061.18, *p* = 0.00) showed that the T index values for all isolates treated with essential oils were significantly different. The post hoc analysis showed that only thyme, *Litsea cubeba* and lemongrass oils produced results that were similar to Funaben T.

Based on the cluster analysis of the sensitivity of individual isolates (Figure 3) and the T index mean values (Table 2), it can be concluded that isolates from the German population are generally more sensitive than isolates from the Polish population. There are exceptions to this rule: *F. poae* FP0D (Germany) is more resistant than *F. culmorum* FCPL (Poland). Furthermore, the isolates of *F. culmorum* and *F. graminearum* were found to be more sensitive than the other three species. Similarly, there are exceptions to this rule: *F. graminearum* FGPL (Poland) is more resistant than *F. poae* FP0D (Germany). Moreover, it seems that the two-species group consisting of *F. culmorum* and *F. graminearum* is more sensitive than the two-species group consisting of *F. avenaceum* and *F. oxysporum*. *F. poae* shows intermediate sensitivity.

Principal component analysis (PCA) for sensitivities (Figure 5) showed that, in terms of T index, the isolates cluster into one relatively large group with three isolates: *F. culmorum* FC2D, *F. graminearum* FG1D and *F. graminearum* FG2D.

They have high values for the second principal component and low values for the first principal component. *F. poae* FP0D can also be included in this group. It reveals slight differences as compared to the group, but its close proximity to the group shows that it is highly similar to the isolates in this group. The remaining isolates are scattered to a greater or lesser extent, reaching different values of both principal components. Note that three *F. culmorum* isolates do not lie next to each other. They occur in three different and distant locations. The distribution of points representing the three isolates of *F. graminearum* is different: two isolates from the German population are grouped together while the third representative (the Polish isolate) is distant from the other two.

### 2.2. Assessment of the Effectiveness of Individual Essential Oils

The combined analysis of the T index values of nine *Fusarium* isolates showed that the mycelium growth is most effectively inhibited by thyme oil (Figure 6).

The activity of thyme oil representing the so-called thymol oils is almost identical to Funaben T and thus can be described as fungicidal. Three other EsO (lemongrass, *Litsea cubeba*, and verbena) clearly exhibit a fungistatic activity, the latter being slightly weaker than the first two. The other three EsO (tea tree, cajeput and grapefruit) have the weakest antifungal activity, of which the latter is weakest. A similar relationship is shown in the cluster analysis diagram (Figure 7).

The groups of bioactive substances with fungicidal and fungistatic activity have not yet been identified. To this end, the correlation matrix of the T index against main groups of compounds was determined (Table 3).

A linear relationship was particularly found between the T index and the presence of monoterpenoids. This is a strong correlation whose sign shows that the T index decreases with the increasing content of monoterpenoids. Monoterpenes and sesquiterpenoids exhibit the average correlation with the T index, with a sign indicating that the T index increases with the increasing content of these groups of compounds. Sesquiterpenes are the only group that does not show any significant effect on the increase of the T index. Due to the high linear dependencies, a multiple regression analysis was performed (Table 4).

The model describing the influence of the main groups of compounds contained in essential oils on the T index clearly shows that all the main groups except sesquiterpenes have a significant effect on the T index. The model accounts for 59% of T index variability. In this model, the T index can be predicted with the error of +/−11.9. Identical data were obtained for the fungistatic activity.

## 3. Discussion

Currently, research is largely focused on selecting oils with high activity and a broad spectrum of activity against microorganisms. Our research focused on the identification of EsO, which at low concentrations exhibit a broad spectrum of fungicidal activity (comparable to fungicides) against parasitic *Fusarium* spp. isolates, regardless of their origin. This is also justified in the light of the more and more frequently observed resistance of *Fusarium* isolates to fungicides and diversified sensitivity to essential oils.

Contemporary publications on *Fusarium* indicate that this morphologically poorly differentiated group of fungi is strongly diverse in genetic terms. The number of chromosomes varies (*n* = 4–20), and their karyotype has core (CCs) and accessory (ACs) chromosomes [24]. Cytological karyotyping shows a variation in chromosome number, also within species. For example, the number of *F. oxysporum* chromosomes varies from *n* = 9–10 to *n* = 19–20 [25].

Within *Fusarium*, a process called horizontal gene transfer (HGT) was identified. HGT is an important mechanism of eukaryotic genome evolution, particularly in unicellular organisms [26]. For example, it was experimentally demonstrated that two LS chromosomes between strains of *F. oxysporum* can be transferred. The transfer leads to the transformation of a non-pathogenic strain into a pathogenic one [27].

*Fusarium* fungi with anamorphic stages (imperfect fungi) do not reproduce sexually. However, they undergo genetic recombination. Recombination occurs during the parasexual cycle. The parasexual cycle involves crossing over [28,29]. Genetic diversity was found in mitochondrial populations present in *Fusarium* cells [30]. The mitochondrial genetic recombination was also reported [31]. Other detected phenomena resulting from the genetic structure of *Fusarium* include the interspecies transfer of chromosomal markers of microsatellites (in *Fusarium oxysporum*) [32] and interspecies variation of ribosomes [33]. The genetic diversity of organisms is significantly increased by mutations. It was found that *Fusarium* is also subject to mutagenesis [34,35].

For *Fusarium* ssp., genetic differentiation of the population structure of races was reported. In particular, the occurrence of specific pathotypes was found [34,35,36,37]. Another paper reported the biological, physiological, and pathogenic differentiation of a genetically homogeneous population of *F. oxysporum* f.sp. cubense. In particular, the differential growth of isolates was observed [38].

Given the occurrence of so many complex genetic mechanisms in *Fusarium* fungi, it is hard to argue with what Professor Shay Covo (Hebrew University, Rechowot, Israel), said about the state of knowledge on fungal pathogens: “Research into the genomic dynamics of fungal plant pathogens is in its infancy” [39]. It is also difficult to argue with the statement of Professor Sephra N. Rampersad: “There is an urgency to supplant the heavy reliance on chemical control of *Fusarium* diseases in different economically important, staple food crops due to development of resistance in the pathogen population, the high cost of production to the risk-averse grower, and the concomitant environmental impacts” [40].

These works indicate the existence of multiple sources of genetic and epigenetic diversity in *Fusarium* spp. In the experiment discussed in this paper, the differences in the mycelial growth rate index of individual isolates were expected and demonstrated. This proves the differential sensitivity of the nine tested isolates to seven commercial EsO. The experiment tested isolates of five species: *F. avenaceum*, *F. culmorum*, *F. graminearum*, *F. oxysporum* and *F. poae*. It was shown that *F. culmorum* and *F. graminearum* isolates, irrespective of origin, were more sensitive to the essential oils than isolates of the other three species. Phylogenetically, *F. culmorum* is similar to *F. graminearum*, both belonging to section Discolour [41].

*F. oxysporum* was most resistant. *Fusarium poae* and *F. avenaceum* showed intermediate properties. *F. avenaceum* was more similar to *F. oxysporum* (the most resistant species). The literature shows that detailed comparative analysis of the mitogenome may offer new insights into the biology of the studied organism and will allow an understanding of the mechanism of sensitivity to essential oils. Mitochondrial genomes are highly informative for resolving phylogenetic relationships even between closely related species and populations. Complete mitochondrial genome sequences offer a stable basis and reference point for phylogenetic and population genetic studies [31].

Our study showed that the right EsO used at low concentrations (up to 2.0%) gives a sound fungicidal effect. Both thyme oil (thymol) and three citral oils (*Litsea cubeba*, lemongrass, and verbena) revealed a strong fungicidal activity, similar to the fungicidal activity of the synthetic fungicide Funaben T confirmed in our previous study [1]. Many researchers emphasise that compounds with phenolic structure (thymol and carvacrol) and terpenoids (citral) are definitely the most effective active ingredients against most fungal species [1,42,43,44,45,46,47]. Due to their lipophilic nature and low molecular weight, these compounds can cause structural and functional damage in the cells of organisms by disrupting the membrane permeability and osmotic balance of the cell, inhibiting the activity of certain enzymes and interfering with ergosterol biosynthesis [48,49,50,51].

Thymol and its isomer carvacrol are definitely the most effective active ingredients against most species of *Fusarium* [52,53,54]. Oils of this type are found not only in *Thymus vulgaris* but also in *Elsholtzia polystachya*, *Origanum vulgare*, *Origanum majorana*, *Citrus limon*, *Coriandrum sativum*, *Trachyspermum ammi*, *Monarda punctata*, *Satureja montana*, *Lavandula multifida*, *Anabasis setifera*, *Zataria multiflora*, *Oliveria decumbens* Vent. [22,43,54,55,56,57,58]. Oils rich in other compounds with phenolic structure—eugenol (*Pimenta dioica* L. clove, cinnamon, allspice, and basil) also show strong fungicidal activity against *Fusarium* [11,18,56,59,60,61].

The citral chemotype includes oil-producing plants whose chemical composition shows a predominance of the monoterpenoid citral (i.e., isomeric mixture of geranial and neral and citronellol), with an admixture of monoterpene hydrocarbons, e.g., myrcene [62]. Citral oils are found in a wide variety of plants, such as *Cymbopogon citratus*, *Verbena officinalis*, *Litsea cubeba*, *Melissa officinalis*, *Aloysia citrodora*, *Vepris macrophylla*, *Citrus bergamia*, *Zingiber officinale*, *Eucalyptus citriodora*, *Salvia officinalis*, *Ocimum gratissimum*, *Lindera citriodora*, *Calypranthes parriculata*, *Tagetes patula*, bitter orange leaves (petitgrain), and lemon peel [20,63,64].

The total inhibition of *Fusarium* mycelial growth is also caused by oils other than thymol and citral, e.g., geranium oil (citronellol and geraniol) [18,60,61]. Similar, strong fungicidal activity, despite differences in chemical composition, is shown in rose oil (linalool) [18,65]. Linalool is also found in coriander, clary sage, lavender oil, and lavandin oil.

These EsO, if used at low concentrations, show the best fungicidal activity against *Fusarium* fungi. This means that they can be used in the development of biodegradable and non-accumulating chemicals (the so-called “green chemicals”).

Three other EsO (cajeput, TTO, and grapefruit) containing active substances other than thymol and citral had a weaker activity on the *Fusarium* isolates. In the presence of the latter, the mycelial growth rate index was similar to the control. Grapefruit oil, with monoterpene (limonene) as the main ingredient, exhibited the weakest activity against the *Fusarium* fungi. This was also confirmed by Thielmann and Muranyi [66]. Other oils with limonene as the main ingredient (lemon oil, tangerine oil, orange oil, and pepper oil) also show low effectiveness at low concentrations [56,67].

EsO with 1,8-cyneol (eucalyptol) as the main ingredient (eucalyptus oil, rosemary oil, laurel oil, turmeric oil, and lavender oil) or α-terpineol (cajeput oil), or its isomer 1-terpinen-4-ol (TTO) also show low effectiveness against the *Fusarium* fungi [67]. In our study, cajeput oil and TTO at low concentrations showed weak activity on the *Fusarium* fungi. However, high fungicidal activity of TTO was also reported [68]. Although some of the results are debatable and the researchers disagree on the issue, biopreparations based on TTO and grapefruit extract essential oils are produced and used. The preparations show long-lasting inhibitory activity against many species of *Fusarium*: *F. avenaceum, F. culmorum, F. graminearum, F. oxysporum*, and *F. poae* [69].

Adequate use of the allelopathic potential of EsO against polyphagous fungi of the *Fusarium* would be safe for humans and the environment [70] and would result in the reduced use of chemical pesticides, contributing to the development of integrated agricultural production [71,72,73]. However, for the selection of oils to be used in green chemicals, the criteria to be taken into account are as follows: the varying sensitivity of *Fusarium* fungi (both fungi within species and isolates belonging to the same species) and the chemical composition of EsO depending on the plant chemotype [11,18,74,75,76].

## 4. Materials and Methods

The study was conducted in two stages. At each stage the fungicidal activity of seven commercial EsO of varying chemical composition (Table 5) against *Fusarium* spp. was evaluated using the disc plate method (method of poisoned substrates) [77,78]:Cultures of fungi were grown in PDA medium for 14 days at 25 °CInoculum. The spore suspension of *Fusarium* spp. in 0.01% sterile Tween 80 were obtained from 14 days old culture. The haemocytometer Thoma was used to obtain a spore suspension of 2 × 10^6^ CFU·cm^3^. Petri dishes (9 cm diameter) containing 20 × cm^3^ PDA medium were inoculating this spore suspension and stored at 25 °C for 14 days. Inoculum—rings with a diameter of 10 mm overgrown by mycelium.Inoculum was placed on the surface of the oil-modified PDA medium.The samples were incubated at 25 °C. Every 2 days, the diameter of developing colonies was measured until the surface of the medium in the control plates was overgrown. Tests were performed in four repetitions (*n* = 4). One petri dish with inoculum (disc overgrown with pathogen mycelium) was treated as a repetitionPDA medium with the Funaben T (at concentrations of 0.125; 0.25 and 0.50%) was used as a positive control. Unmodified PDA medium (without oils) with a ring was used as a negative control

The tests were performed in four repetitions (*n* = 4), taking as a repetition one Petri dish from the inoculum in the form of a disc overgrown with pathogen mycelium.

In the study were used EsO, i.e., thyme (T), *Thymus vulgaris* (produced by MELASAN, Eugendorf, Austria); lemongrass (L), *Cymbopogon citratus* (Lemongrass), *Litsea cubeba* (LC), *Litsea cubeba,* and grapefruit (G), *Citrus paradisi* (produced by TAOASIS GmbH, Berlin, Germany); verbena (V), *Lippia javanica* (produced by Piping Rock Health Products, LLC, Ronkonkoma, NY 11,779 USA); tea tree (TTO), *Melaleuca alternifolia* (produced by MEDESIGN IC GmbH Dietramszell—Linden, Germany); cajeput (C), *Melaleuca leucadendron var. cajaputi* (produced by PRIMAVERA LIFE GmbH, Oy-Mittelberg, Germany. Based on the chemical composition, the following EsO groups were distinguished: three citral oils (lemongrass, *Litsea cubeba*, and verbena), one thymol oil (thyme oil), two oils containing mainly monocyclic monoterpenoids, i.e., 1-terpinen-4-ol (tea tree oil (TTO) and α-terpineol (cajeput oil), and one limonene (grapefruit oil). The following concentrations of EsO were used: 0.025; 0.05; 0.125; 0.25; 0.50; 1.0; and 2.0%.

The oil colloid solutions were prepared in water with 0.05% Tween 80 (produced by BTL, Poland) and fed into a liquefied PDA medium (Potato Dextrose Agar (BIOCORP, Warszawa, Poland). A relative control of the effectiveness of EsO was chemical seed treatment Funaben T (containing 20% carbendazim and 45% thiocarbamate), produced by Zakłady Chemiczne “Organika Azot” S.A., Jaworzno, Poland), applied in concentrations lower, higher and recommended by the manufacturer (0.125, 0.25, and 0.5%).

At the first stage, the fungistatic activity of EsO was evaluated against four species of *Fusarium* from the Polish population (*F. avenaceum* FAPL, *F. culmorum* FCPL, *F. graminearum* FGPL and *F. oxysporum* FOPL) isolated from infected wheat kernels in south-west Poland in 2012–2014 [1]. The fungistatic activity of the tested oils was evaluated on the basis of the percentage of inhibition of fungal colony growth calculated from the Abbott formula [1]. The correlation coefficient between the mycelial growth rate index (T) and the fungistatic activity (FA) was determined. This relationship was expressed by the formula:FA = 99.74 − 2.00 ∗ T(1)

At the second stage, on the basis of the mycelium growth rate index, the effectiveness of the EsO in limiting the rise of *Fusarium* isolates from the German population (*F. culmorum* FC1D, *F. culmorum* FC2D, *F. graminearum* FG1D, *F. graminearum* FG2D and *F. poae* FP0D) was assessed. The isolates were separated in 2012–2014 from infected wheat kernels obtained from Leibniz Zentrum für Agrarlandschaftsforschung e.V., Institut für Landschaftsbiogeochemie (ZALF, Müncheberg, Germany) collection.

The fungicidal activity of seven EsO (with concentrations of 0.025, 0.05, 0.125, 0.25, 0.50, 1.0 and 2.0%) against five *Fusarium* isolates from the German population was evaluated too, using the disc plate method.

The growth rate index (T) of the isolates was determined based on measurements of mycelial colony growth using the formula:(2)$$T=+\frac{b_{1}}{d_{1}}+\cdots \frac{b_{x}}{d_{x}}$$
where T—growth rate index; A—average measurement value of diameter colonies [mm]; D—duration of the experiment; *b*_1_ (…) *b*_x_—increase in colonies diameter [mm]; *d*_1_ (…) *d*_x_—number of days since last measurement.

The results of the first and second stage presented as a growth rate index were then used to indicate the EsO (as potential natural fungicides) effectively limiting the development of various common Central European parasitic species *Fusarium* spp. Finally, the oils sensitivity of four *Fusarium* isolates from the Polish population and five *Fusarium* isolates from the German population were compared. In order to standardize the description of isolates, the relevant markings were introduced. Polish population (symbols used in the cited publication are given in brackets): *F. avenaceum* FAPL (GM2), *F. culmorum* FCPL (KP17), *F. graminearum* FGPL (L22) and *F. oxysporum* FOPL (P6). German population (symbols used in ZALF are given in brackets): *F. culmorum* FC1D (ZALF 186), *F. culmorum* FC2D (ZALF 187), *F. graminearum* FG1D (ZALF 24), *F. graminearum* FG2D (ZALF 339) and *F. poae* FP0D (ZALF 338). All tested isolates were stored on PDA slants at 4 °C and subcultured every two months.

### Statistical Data Analysis

Statistical analysis of the mycelial growth rate index (T) in the presence of each of the seven EsO at different concentrations was performed. Each experimental variant was repeated four times for nine isolates. For each experimental variant, the values of descriptive statistics (mean, median, minimum value, and maximum value) were determined. The Shapiro–Wilk test was used to check whether the mycelial growth rate index (in the presence of a certain amount of EsO) is a variable of normal distribution. Next, a non-parametric Kruskal–Wallis test was applied to assess whether the mycelial growth rate index (T) of *Fusarium* isolates in the presence of individual EsO and the activity of Funaben T differ from the control test for each of the mycelium isolates separately. If the test results were significant, multiple comparison of mean ranks for all groups (the post hoc analysis) was used to determine which pairs of essential oils differ from each other.

Additionally, the relationship between the percentage share of a given group of compounds in the EsO and the growth rate index of a given isolate in the presence of a certain amount of essential oil was examined. For this purpose, the chemical compounds contained in the EsO were divided into monoterpenes, terpenoids, sesquiterpenes, sesquiterpenoids, and other compounds, and a distinction was made between the main groups of compounds. Next, the correlation coefficients between the growth rate index of a given isolate in the presence of a certain amount of EsO and the percentage of a given group of compounds in the oil were determined.

The correlation coefficient between the growth rate index (T) and the fungistatic activity (FA) was determined. Since the correlation coefficient was high, the linear regression equation describing the relationship between FA and T was determined.

Cluster analysis and principal component analysis (PCA) were used to group the oils in terms of their effectiveness on individual isolates and to find similarities in the response of individual isolates to the oils. All statistical analyses were performed using STATISTICA 13.0 (StatSoft, Inc. TIBCO Software Inc., Carlsbad, CA, USA) at the significance level of 0.05.

## 5. Conclusions

A growing body of evidence shows that essential oils significantly reduce the growth of *Fusarium* spp. and minimize the risk of pathogens acquiring resistance. Moreover, the oils are characterized by low toxicity to humans and the environment, as they are biodegradable. Our research focused on the identification of EsO, which at low concentrations exhibit a broad spectrum of fungicidal activity (comparable to fungicides) against parasitic *Fusarium* spp. isolates, regardless of their origin. Given the diverse sensitivity of *Fusarium* spp., it is difficult to choose the type and concentration of such an oil. The sensitivity of individual *Fusarium* species varied. Their sensitivity, regardless of the isolates origin, in order from the most to the least sensitive, is as follows: *F. culmorum*, *F. graminearum*, *F. poae, F. avenaceum and F. oxysporum*. *Fusarium* isolates from the German population (*F. culmorum* FC1D, *F. culmorum* FC2D, *F. graminearum* FG1D, *F. graminearum* FG2D and *F. poae* FP0D) were generally more sensitive than those from the Polish population (*F. avenaceum* FAPL, *F. culmorum* FCPL, *F. graminearum* FGPL and *F. oxysporum* FOPL). Thyme oil has also shown a concentration independent fungicidal effect (similar to Funaben T). Citral oils (lemongrass and *Litsea cubeba*) acted in a similar way, but in a concentration above 0.025%. On the other hand, the fungicidal activity of the remaining oils (cajeput, verbene, TTO and grapefruit) depended on the concentration and sensitivity of the tested *Fusarium* isolate. Our observations demonstrate that the construction of “green chemicals” should focus on thymol and citral oils. Therefore, the presented research results may contribute to the effective protection of plants in agro-ecosystems. On the other hand, a detailed comparative analysis of the mitogenome of *Fusarium* spp. may offer new insights into the biology of the studied organism and will allow an understanding of the mechanism of sensitivity to essential oils.

## Figures and Tables

**Figure 1 molecules-26-06488-f001:**
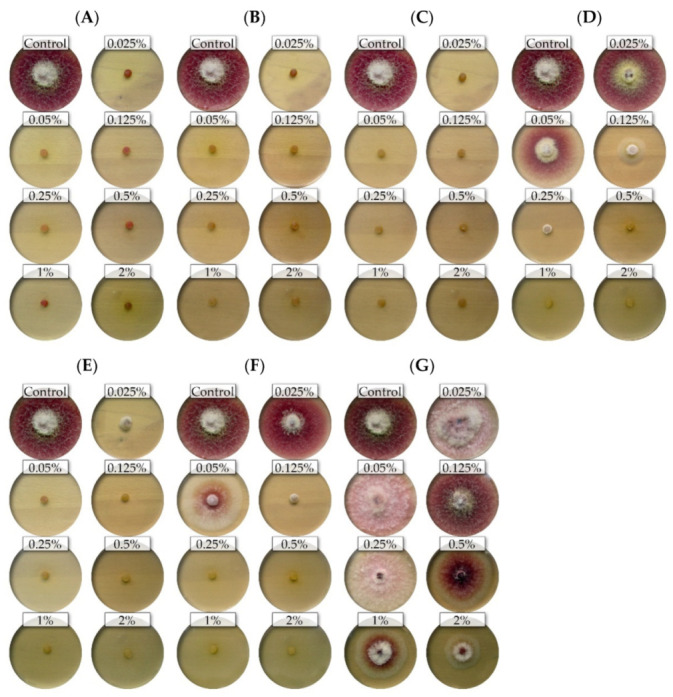
The effects of the different concentrations of the essential oils on the mycelial growth of the *Fusarium culmorum* FC2D: (**A**)—thyme (T); (**B**)—lemongrass (L); (**C**)—*Litsea cubeba* (LC); (**D**)—cajeput (C); (**E**)—verbena (V), (**F**)—TTO; (**G**)—grapefruit (G) (light background).

**Figure 2 molecules-26-06488-f002:**
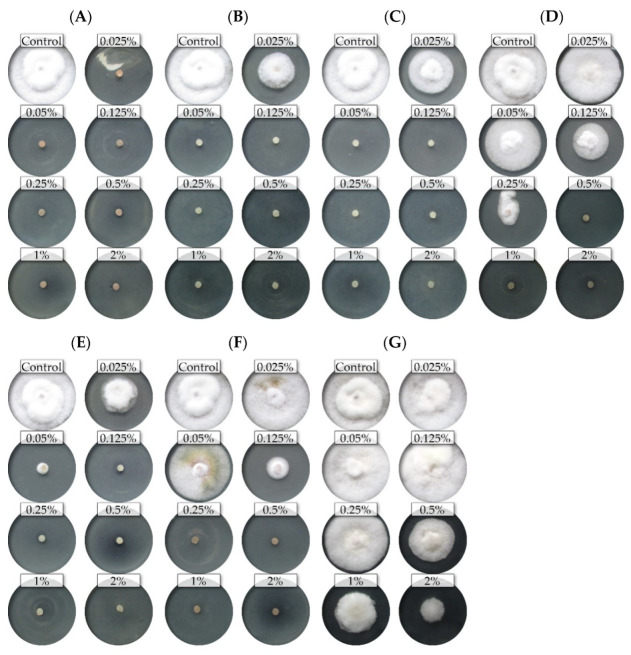
The effects of the different concentrations of the essential oils on the mycelial growth of the *Fusarium poae* FP0D: (**A**)—thyme (T); (**B**)—lemongrass (L); (**C**)—*Litsea cubeba* (LC); (**D**)—cajeput (C); (**E**)—verbena (V), (**F**)—TTO; (**G**)—grapefruit (G) (dark background).

**Figure 3 molecules-26-06488-f003:**
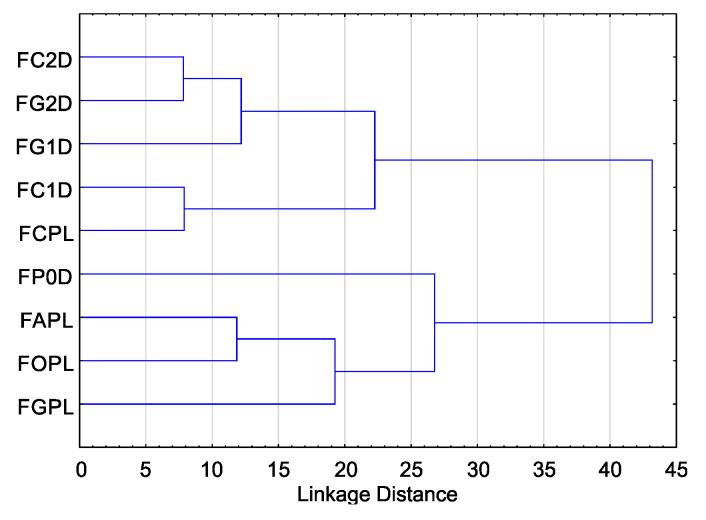
Variability of sensitivity of *Fusarium* spp. isolates—ordered after cluster analysis (D-Germany, PL-Poland.): FC2D—*F. culmorum*, FG2D—*F. graminearum*, FG1D—*F. graminearum*, FC1D—*F. culmorum*, FCPL—*F. culmorum*, FP0D—*F. poae*, FAPL—*F. avenaceum*, FOPL—*F. oxysporum*, FGPL—*F. graminearum*.

**Figure 4 molecules-26-06488-f004:**
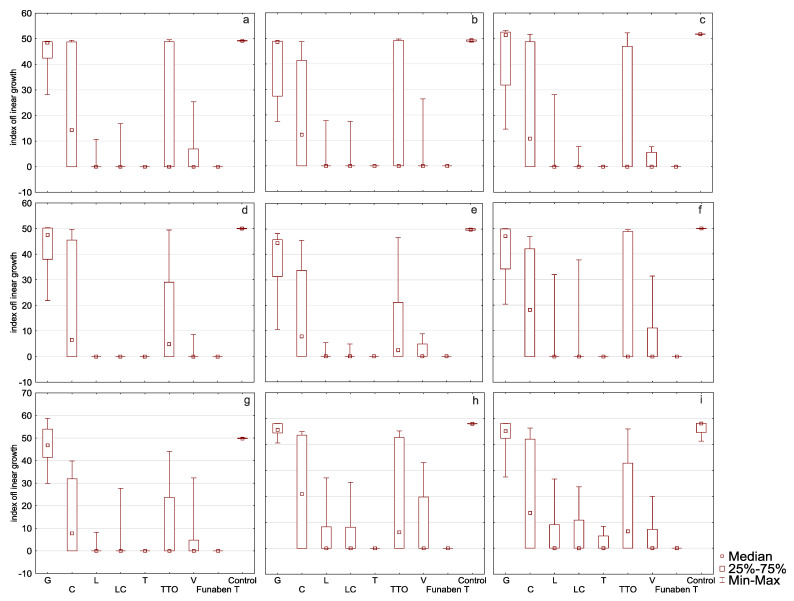
Indexes of linear growth (T) of *Fusarium* ssp. isolates for all concentration of EsO together (D-Germany, PL-Poland): (**a**) FC2D—*F. culmorum*, (**b**) FG2D—*F. graminearum*, (**c**) FG1D—*F. graminearum*, (**d**) FC1D—*F. culmorum*, (**e**) FCPL—*F. culmorum*, (**f**) FP0D—*F. poae*, (**g**) FGPL—*F. graminearum*, (**h**) FAPL—*F. avenaceum*, (**i**) FOPL—*F. oxysporum*; index T calculated after treated of isolates with essential oils and Funaben T: G—grapefruit, C—cajeput, L—lemongrass, LC—*Litsea cubeba*, T—thyme, TTO—tea tree, V—verbena. Control-isolates without EsO or Funaben T.

**Figure 5 molecules-26-06488-f005:**
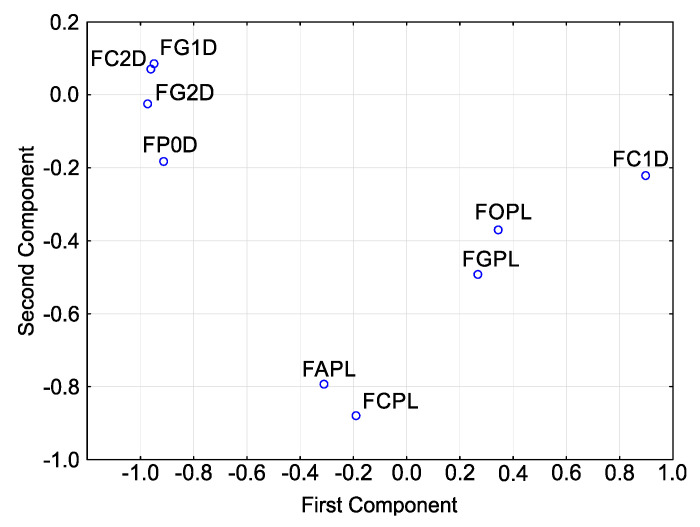
Variability of sensitivity of *Fusarium* spp. isolates—ordered after cluster analysis (D-Germany, PL-Poland.): FC2D—*F. culmorum*, FG2D—*F. graminearum*, FG1D—*F. graminearum*, FC1D—*F. culmorum*, FCPL—*F. culmorum*, FP0D—*F. poae*, FAPL—*F. avenaceum*, FOPL—*F. oxysporum*, FGPL—*F. graminearum*.

**Figure 6 molecules-26-06488-f006:**
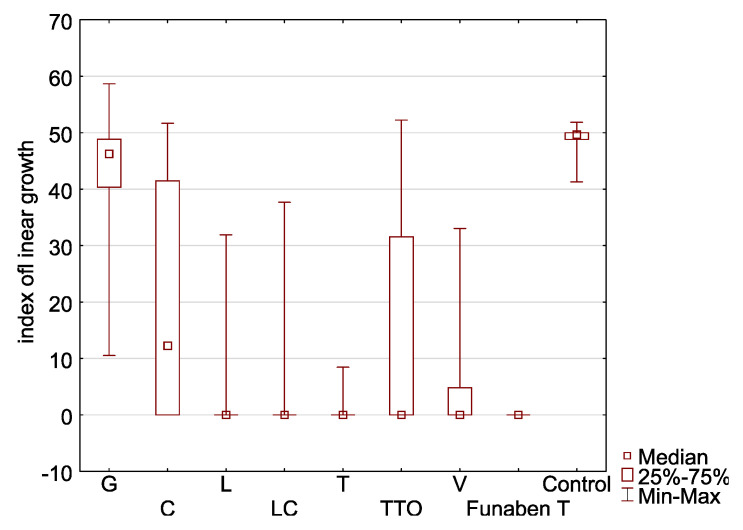
Characteristic of the indexes of linear growth (T) of *Fusarium* spp. isolates (aggregate numbers of all tested isolates) in relation to the used essential oils and Funaben T: G—grapefruit, C—cajeput, L—lemongrass, LC—*Litsea cubeba*, T—thyme, TTO—tea tree, V—verbena.

**Figure 7 molecules-26-06488-f007:**
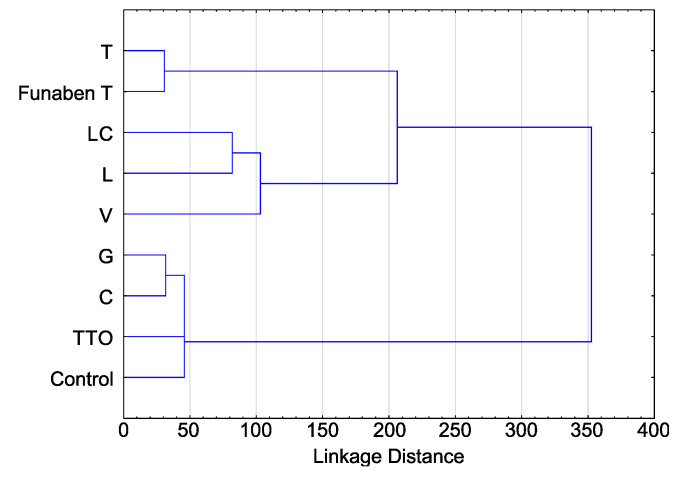
The clustering of the indexes of linear growth (T) of *Fusarium* spp. isolates (aggregate numbers of all tested isolates) in relation to the used essential oils and Funaben T: C—cajeput, G—grapefruit, L—lemongrass, LC—*Litsea cubeba*, T—thyme, TTO—tea tree, V—verbena.

**Table 1 molecules-26-06488-t001:** Results of Kruskal–Wallis test. Comparison of the index of linear growth of mycelial isolate in presence of used essential oils.

Isolates	H (8, 212)	*p*
FAPL	124.68	0.00 *
FCPL	123.34	0.00 *
FC1D	145.2	0.00 *
FC2D	114.44	0.00 *
FGPL	131.64	0.00 *
FG1D	120.18	0.00 *
FG2D	115.97	0.00 *
FOPL	111.1	0.00 *
FP0D	111.5	0.00 *

*—the results statistically significant.

**Table 2 molecules-26-06488-t002:** Descriptive statistics for the growth rate index of all tested *Fusarium* isolates; for all concentrations of essential oils together *Fusarium*: FC2D—*F. culmorum*, FG2D—*F. graminearum*, FG1D—*F. graminearum*, FC1D—*F. culmorum*, FCPL—*F. culmorum*, FP0D—*F. poae*, FGPL—*F. graminearum*, FAPL—*F. avenaceum*, FOPL—*F. oxysporum*. control–isolates without EsO or Funaben T. Notation: N—test volume, SD—standard deviation.

Oil	Isolate	N	Mean	SD	Isolate	N	Mean	SD	Isolate	N	Mean	SD
Grapefruit		28	44.57	6.72		28	43.13	8.98		28	46.87	8.04
Cajeput		28	22.52	22.82		28	17.00	20.65		27	15.75	16.13
Lemongrass		28	0.75	2.37		28	0.00	0.00		28	1.06	2.68
*Litsea cubeba*		28	1.48	4.00		28	0.00	0.00		28	1.71	5.53
Thyme	FC2D	28	0.00	0.00	FC1D	28	0.00	0.00	FGPL	29	0.00	0.00
Tea tree		28	16.17	21.85		28	13.31	18.69		28	10.97	15.52
Verbena		28	3.62	6.66		28	0.98	2.49		28	4.04	8.55
Control		12	49.18	0.13		12	50.02	0.11		12	49.86	0.19
Funaben T		4	0.00	0.00		4	0.00	0.00		4	0.00	0.00
Grapefruit		28	40.51	11.98		27	38.19	11.65		28	45.81	2.4
Cajeput		28	20.36	20.88		28	15.06	17.71		27	21.06	18.65
Lemongrass		28	2.44	6.10		28	0.76	1.89		28	4.65	8.55
*Litsea cubeba*		28	2.42	6.04		28	0.68	1.70		28	4.66	8.54
Thyme	FG2D	28	0.00	0.00	FCPL	27	0.00	0.00	FAPL	28	0.00	0.00
Tea tree		28	15.76	22.04		28	11.32	16.57		28	17.12	19.38
Verbena		28	3.73	9.31		28	1.79	3.00		28	6.8	11.19
Control		12	49.15	0.40		12	49.83	0.33		12	48.03	0.09
Funaben T		4	0.00	0.00		4	00.00	0.00		4	00.00	0.00
Grapefruit		28	42.59	13.22		28	40.69	10.43		28	43.52	6.34
cajeput		28	21.09	22.36		28	19.58	19.22		28	19.75	18.75
Lemongrass		28	2.96	8.08		28	4.5	11.22		28	5.09	9.06
*Litsea cubeba*		28	1.10	2.74		28	5.36	13.36		28	4.71	8.07
Thyme	FG1D	28	0.00	0.00	FP0D	28	0.00	0.00	FOPL	28	1.77	2.94
Tea tree		28	16.3	22.5		28	16.79	21.84		28	13.9	17.36
Verbena		28	1.90	3.10		28	6.02	11.09		28	3.1	5.39
Control		12	51.72	0.08		12	49.93	0.07		12	46.39	3.41
Funaben T		4	0.00	0.00		4	0.00	0.00		4	0.00	0.00

**Table 3 molecules-26-06488-t003:** Growth rate index correlation matrix against the main groups of compounds in essential oils; the obtained results showed no statistical significance.

Monoterpenes	Monoterpenoids	Sesquiterpenes	Sesquiterpenoids	Other Chemical Compounds
0.41	−0.64	−0.03	0.47	0.66

**Table 4 molecules-26-06488-t004:** The model of multiple regression: b—regression coefficient; b^1^—standardized coefficient; SE standard error; *t*—Student’s *t*-test value.

	b^1^	SE of b^1^	b	SE of b	*t* (1754)	*p*
Intercept			736.24	73.69	9.99	0.00 *
Monoterpenes	−5.42	0.56	−6.87	0.71	−9.62	0.00 *
Monoterpenoids	−8.49	0.85	−7.55	0.76	−9.94	0.00 *
Sesquiterpenes	−1.46	0.13	−6.65	0.59	−11.09	0.00 *
Sesquiterpenoids	0.02	0.06	0.45	1.19	0.38	0.70
Other chemical compounds	−4.24	0.45	−7.08	0.76	−9.32	0.00 *
Concentration	−0.21	0.016	−6.21	0.46	−13.40	0.00 *

*—the results statistically significant.

**Table 5 molecules-26-06488-t005:** Chemical composition of the tested essential oils in [%]: T—thyme; L—lemongrass; LC—*Litsea cubeba*; V—verbena; TTO—tea tree; C—cajeput; G—grapefruit [1].

Compound	RI		Etheric Oils						
	Lit *	Cal *	T	L	LC	V	TTO	C	G
Monoterpenes									
Tricyclene	923	920	0.17 ± 0.01	0.44 ± 0.08		0	0		
α-thujene	928	928				0.44 ± 0.05	0.83 ± 0.07		
α-pinene	936	933	2.75 ± 0.09	0.49 ± 0.11	2.86 ± 0.16	0	3.42 ± 0.06	5.37 ± 0.01	3.27 ± 0.01
Camphene	950	947	1.93 ± 0.07	3.71 ± 0.06	0.58 ± 0.04	0.80 ± 0.02	0		
β-pinene	978	974	0.65 ± 0.02	0	3.95 ± 0.08	1.08 ± 0.05	0.81 ± 0.02	3.93 ± 0.15	
β-myrcene	989	991	2.44 ± 0.03				0.38 ± 0.11	3.01 ± 0.07	5.32 ± 0.01
α-phellandrene	1004	1002	0.87 ± 0.03			0.05 ± 0.02	0.15 ± 0.08		
Sabinene (4,10-thujene)	1004	1009				0.27 ± 0.04	0.17 ± 0.03		1.56 ± 0.03
3-carene	1011	1005					17.04 ± 0.15		
α-terpinene	1017	1018	2.32 ± 0.10				10.29 ± 0.09		
p-cymene	1024	1020				3.62 ± 0.03			
Limonene	1029	1026	15.15 ± 0.18		20.94 ± 0.13				34.63 ± 0.73
γ-terpinene	1060	1061	8.10 ± 0.07			2.02 ± 0.07		0.37 ± 0.04	0.78 ± 0.01
Terpinolene	1087	1087				0.45 ± 0.01	3.87 ± 0.05	0.27 ± 0.02	0.08 ± 0.01
β-patchulene	1457	1455					0.16 ± 0.04		
Sum monoterpenes			34.38	4.64	28.33	8.73	37.12	12.95	45.64
Monoterpenoids									
α and β citral (geranial and neral)	-	-		68.94 ± 0.10	61.72 ± 0.43	36.00 ± 0.08			
Trifluorolavandulol		1999							2.19 ± 0.07
Eucalyptol	1031	1027				13.46 ± 0.17	13.90 ± 0.15	18.50 ± 0.05	
Linalool oxide	1065	1064						0.12 ± 0.03	
Linalool	1099	1105	8.90 ± 0.18	5.73 ± 0.22	2.58 ± 0.04	8.53 ± 0.01		11.19 ± 0.17	4.83 ± 0.039
1-terpineol	1137	1135	1.19 ± 0.05		1.39 ± 0.06	0.47 ± 0.01		0.87 ± 0.10	
p-menth-3-en-9-ol	1141	1140		0.71 ± 0.02					
Camphor	1143	1141				4.62 ± 0.03			
Verbenol	1145	1145							0.18 ± 0.014
β-citronellal	1154	1152			1.87 ± 0.13				0.42 ± 0.0.03
Borneol	1166	1168	3.07 ± 0.09	2.93 ± 0.07		1.32 ± 0.02			
1-terpinen-4-ol	1177	1181				4.51 ± 0.05	38.24 ± 0.38	4.41 ± 0.35	0.09 ± 0.003
α-terpineol	1190	1197	1.14 ± 0.09		1.02 ± 0.06	18.26 ± 150	6.88 ± 0.04	36.57 ± 0.21	1.83 ± 0.048
α-pinene oxide	1197	1195							0.51 ± 0.029
cis-geraniol	1238	1234							0.55 ± 0.046
β citral (neral)	1242	1231							0.92 ± 0.058
trans-geraniol	1255	1252							0.45 ± 0.04
Linalyl acetate	1255	1260	0.93 ± 0.06						1.87 ± 0.028
Geranial	1270	1269							1.36 ± 0.022
Thymol	1290	1298	45.75 ± 0.18						
α-terpinyl acetate	1347								0.23 ± 0.021
Nerol acetate	1363	1366		1.68 ± 0.01					
Geraniol acetate	1380	1385							2.26 ± 0.045
Sum momoterpenoids			60.98	79.99	68.58	87.18	59.02	71.66	17.69
Sesquiterpenes									
α-cubebene	1351	1350						0.52 ± 0.04	0.37 ± 0.004
α-longipinene	1352	1350		0.67 ± 0.08					
ylangene	1370	1370						0.51 ± 0.01	
β-cubebene	1387	1390							0.49 ± 0.028
β-elemene	1388	1387						0.14 ± 0.05	
Longifolene	1407	1408				1.12 ± 0.02			
α-gurjunene	1409	1410					0.23 ± 0.02	1.19 ± 0.09	
caryophyllene	1419	1423	4.31 ± 0.02	3.76 ± 0.012	2.45 ± 0.018	0.55 ± 0.06	0.28 ± 0.003	2.60 ± 0.19	0.98 ± 0.061
α-caryophyllene	1420	1408	0.33 ± 0.03	0.45 ± 0.01	0.20 ± 0.004			1.70 ± 0.03	0.14 ± 0.014
β-gurjunene	1431	1430		1.15 ± 0.05			1.23 ± 0.05	0.57 ± 0.03	
(+)aromadendrene	1441	1440					0.94 ± 0.10		
γ-elemene	1449	1445						0.05 ± 0.01	
Allo-aromadendrene	1460	1458					0.23 ± 0.03	3.63 ± 0.06	
γ-muurolene	1476	1478					0.12 ± 0.06		
Germacene D	1481	1496						0	0.18 ± 0.01
(+)-valencene	1491	1499		0					0.14 ± 0.09
β-selinene	1493	1490						1.62 ± 0.03	
γ-cadinene	1513	1517		4.83 ± 0.10					
σ-cadinene	1523	1526					0.83 ± 0.04		0.48 ± 0.009
Cadinene	1533	1530						0.37 ± 0.05	
Sum sesquiterpenes			4.64	10.86	2.65	1.67	3.86	12.90	2.83
Sesquiterpenoids									
trans-nerolidol	1524	1522							0.02 ± 0.006
elemol	1536	1540							0.05 ± 0.01
Caryophyllene oxide	1581	1572		1.14 ± 0.03	0.44 ± 0.05			0.42 ± 0.08	0.23 ± 0.003
Guaiol	1589	1590						0.55 ± 0.05	
Eudesmol	1616	1611						1.52 ± 0.03	
Farnesol	1722	1718							0.05 ± 0.011
Nootkatone	1813	1818							1.37 ± 0.069
Farnesyl acetate	1818	1820							0.03 ± 0.002
Sum sesquiterpenoids				1.14	0.44			2.49	1.75
Sum other chemical compounds				3.37					26.81

Lit *—Literature values of Kovats retention indexes [79]. Cal *—The average value of the relative composition of the essential oil percentage was calculated from the peak areas.

## Data Availability

Not applicable.

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
