# Peer review of "Effectiveness of the Influence of Selected Essential Oils on the Growth of Parasitic Fusarium Isolated from Wheat Kernels from Central Europe"

_molecules, 2021, doi:10.3390/molecules26216488_

Round 1

Reviewer 1 Report

Comments and Suggestions for Authors

The article “Effectiveness of the influence of selected essential oils on the growth of parasitic Fusarium isolated from wheat kernels from Central Europe” contains original and very interesting results. However, the article needs to be improved in the presentation of the text and figures.

In their manuscript, authors very often refer to the previous work of Krzyśko-Łupicka, T .; Sokół, S .; Piekarska-Stachowiak, A. Evaluation of Fungistatic Activity of Eight Selected Essential Oils on Four Heterogeneous Fusarium Isolates Obtained from Cereal Grains in Southern Poland. Molecules 2020, 25, 292; doi: 10.3390 / molecules25020292. It is very uncomfortable. I had to read 2 articles at once.

In this article, the materials and methods section is written briefly and casually. In the Results section, many results are not presented, but statistical analysis has been carried out on them. I recommend that the authors expand the manuscript, describe the techniques in more detail, add tables and decipher some concepts.

My comments are as follows:

  1. Often in the text there are two concepts of fungistatic activity and fungicidal activity. These are different concepts. I recommend that the authors clearly distinguish between which essential oils have a fungistatic effect, and which essential oils have a fungicidal effect. In the text of the manuscript, these concepts are sometimes confused. (Lines 61, 69, 89, 92, 415).
  2. Latin names in many places in the manuscript are not in italics. Authors should correct the spelling of Latin names. (Lines 320 – 326, 330 – 334, 377 – 383, 406).
  3. The abbreviation of species pluralis is indicated incorrectly in Latin. (Lines 24, 143, 283, 297).
  4. In the abstract and methods it is written that various concentrations of EO were used (with concentrations of 0.025; 0.05; 0.125; 0.25; 0.50; 1.0, and 2.0%) (Lines 26 – 27, 384 – 385, 415 - 416), and Funaben T was also used at various concentrations (0.125, 0.25, and 0.5%) (Lines 390, 418). However, it seems to me that only one concentration of EO and Funaben T is presented in Table 2 and Figure 2. The authors should present the results for all concentrations for strains from Germany and refer to previous work by Krzyśko-Łupicka et al. (2020) on strains from Poland. Or the authors should describe in the text of the manuscript how they selected and chose the concentrations with a fungistatic effect, what concentrations were used when presenting the results in Table 2 and Figure 2. In accordance with the previous work, all EOs have, in addition to fungistatic concentrations, fungicidal concentrations, when the mycelium of the fungus completely stops.
  5. I recommend that the authors take a closer look at the Materials and Methods
  6. Lines 366 - 372 should be removed. This is an instruction from the template.
  7. Lines 373 - 414, where there is a description of the strains of pathogens, essential oils and their concentrations, as well as the chemical composition of some oils, I recommend systematizing and presenting them in the table for easier perception.
  8. I would also like to draw the attention of the authors to the chemical composition of essential oils, which they mention. (lines 410 - 414). It is important to describe the chemical composition of each essential oil and make a reference to previous work, since the authors make a statistical analysis of the dependence of the mycelium growth index on the chemical composition of the essential oil.
  9. Lines 415 - 417. The authors use the disk method to measure the fungicidal activity of essential oils. In this case, the authors measure the growth rate of mycelium, whereas in this method, the zone of growth inhibition is traditionally measured. Therefore, I recommend that the authors not refer to previous work, but describe the method in detail so that there is no confusion. In addition, a large amount of data is very difficult to perceive without a visual picture of mycelium growth. I recommend the authors to present a photo of Petri dishes with the growth of fungal mycelium with the addition of various essential oils, several typical examples of the fungicidal and fungistatic effect in several species. (Present in the Results section).
  10. I recommend transferring lines 87 - 93 to materials and methods.
  11. Results section.
  12. Figures 1, 2, 3, 4, 5 should be corrected. Increase the figures themselves and the size of the captions along the X and Y axes.
  13. The cajeput essential oil is denoted as (K) in the methods, and in the figure caption as C. It should be corrected.
  14. Authors should clarify what constitutes a "control" in Table 2 and Figure 2.
  15. Lines 216-221 and lines 226-230 contain repeats. It should be corrected.
  16. Tables 3 and 4 describe the statistical analysis of the results not presented in the manuscript. Authors should imagine the results and their statistical analysis. In other words, the authors should provide a description of the chemical composition of essential oils and relate it to the fungicidal and fungistatic activity of each oil. The authors discuss the chemical composition of essential oils in the Discussion section, but it seems to me that the data on the chemical composition of essential oils presented in the table in the Methods section will improve the understanding of this manuscript.
  17. Section Discussion.
  18. Lines 255 - 258 should be removed. This is an instruction from the template.
  19. The abbreviation for essential oils –EsO and EOs - is used in different ways. It should be corrected.

Author Response

We would like to thank the Reviewer for careful reading of this manuscript and for the comments and suggestions, which help to improve our manuscript. We have tried to do our best to respond to any comments and improve manuscript. As indicated below, we have made changes accordingly to all the general and specific comments. As suggested, the manuscript was expanded and the description of the techniques was refined. We added tables and the terms clarified. Changes and additions to the manuscript (marked in red).

  1. Often in the text there are two concepts of fungistatic activity and fungicidal activity. These are different concepts. I recommend that the authors clearly distinguish between which essential oils have a fungistatic effect, and which essential oils have a fungicidal effect. In the text of the manuscript, these concepts are sometimes confused. (Lines 61, 69, 89, 92, 415). The fungicidal and fungistatic terms have been clarified - we corrected this (now the lines 72-75)
  2. Latin names in many places in the manuscript are not in italics. Authors should correct the spelling of Latin names. (Lines 320 – 326, 330 – 334, 377 – 383, 406) - we corrected this. (lines 336-340; 346-349; 401-407).
  3. The abbreviation of species pluralis is indicated incorrectly in Latin. (Lines 24, 143, 283, 297) - we corrected this (lines 21, 31, 122, 214, 231, 242, 309, 495)
  4. In the abstract and methods it is written that various concentrations of EO were used (with concentrations of 0.025; 0.05; 0.125; 0.25; 0.50; 1.0, and 2.0%) (Lines 26 – 27, 384 – 385, 415 - 416), and Funaben T was also used at various concentrations (0.125, 0.25, and 0.5%) (Lines 390, 418). However, it seems to me that only one concentration of EO and Funaben T is presented in Table 2 and Figure 2. The authors should present the results for all concentrations for strains from Germany and refer to previous work by Krzyśko-Łupicka et al. (2020) on strains from Poland. Or the authors should describe in the text of the manuscript how they selected and chose the concentrations with a fungistatic effect, what concentrations were used when presenting the results in Table 2 and Figure 2. In accordance with the previous work, all EOs have, in addition to fungistatic concentrations, fungicidal concentrations, when the mycelium of the fungus completely stops. In presented studies analyzed seven various concentrations for each essential oils and three concentrations of Funaben T according to description in Material and methods. In Table 2 and Figure 2 we published values of descriptive statistics for all used concentrations of each oil together. In each of the variants, they were four repeats, in the Table 2 for each oil is N = 28 (7 concentration x 4 repetitions). For Funaben T we considered three concentrations. In this case in Table 2 and Figure 2 we published values of descriptive statistics for all concentrations together. In table for Funaben T N=12 (3 concentrations x 4 repetitions). We decided add Table Minimum of growth rate index of the analysed oil at minimum concentration of the isolate in Supplementary Materials for a better explanation (lines 127-134; 160-164)
  5. I recommend that the authors take a closer look at the Materials and Methods section- we corrected this
  6. Lines 366 - 372 should be removed. This is an instruction from the template - this part has been deleted.
  7. Lines 373 - 414, where there is a description of the strains of pathogens, essential oils and their concentrations, as well as the chemical composition of some oils, I recommend systematizing and presenting them in the table for easier perception - in the Materials and Methods section, we added a Table 5: Chemical composition of the tested essential oils in [%]: T—thyme; L—lemongrass; LC—Litsea cubeba; V—verbena; TTO—tea tree; C—cajeput; G—grapefruit [1] (lines 415-417; 411-412)
  8. I would also like to draw the attention of the authors to the chemical composition of essential oils, which they mention. (lines 410 - 414). It is important to describe the chemical composition of each essential oil and make a reference to previous work, since the authors make a statistical analysis of the dependence of the mycelium growth index on the chemical composition of the essential oil - we added a table 5
  9. Lines 415 - 417. The authors use the disk method to measure the fungicidal activity of essential oils. In this case, the authors measure the growth rate of mycelium, whereas in this method, the zone of growth inhibition is traditionally measured. Therefore, I recommend that the authors not refer to previous work, but describe the method in detail so that there is no confusion. In addition, a large amount of data is very difficult to perceive without a visual picture of mycelium growth. I recommend the authors to present a photo of Petri dishes with the growth of fungal mycelium with the addition of various essential oils, several typical examples of the fungicidal and fungistatic effect in several species. (Present in the Results section) - in the Materials and Methods section we added a detailed description of the method of the performed determination (lines 382-400). In the Results section, we added an example photo of Petri dishes with mycelium development in the presence of various concentrations of the tested essential oils

- Figure1. The effects of the different concentrations of the essential oils on the mycelial growth of the Fusarium culmorum FC2D: A - thyme (T); B - lemongrass (L); C - Litsea cubeba (LC); D - cajeput (C); E - verbena (V), F - TTO; G - grapefruit (G) (light background. (lines 98-104)

- Figure 2. F. poae FP0D mycelium growth inhibition zones in PDA medium after 14-day incubation in the presence of EsO, at concentrations ranging from 0.025% to 2.0%: A - thyme (T); B - lemongrass (L); C - Litsea cubeba (LC); D - cajeput (C); E - verbena (V), F - TTO; G - grapefruit (G). (lines 105-106)

  1. I recommend transferring lines 87 - 93 to materials and methods - we corrected this. (lines 429 - 430)

  1. Results section.
  2. Figures 1, 2, 3, 4, 5 should be corrected. Increase the figures themselves and the size of the captions along the X and Y axes - we corrected this (lines 120, 159, 213, 230, 241)
  3. The cajeput essential oil is denoted as (K) in the methods, and in the figure caption as C. It should be corrected.- we corrected this (line 406)
  4. Authors should clarify what constitutes a "control" in Table 2 and Figure 2 - In this manuscript term ‘control” means that the isolates developed without any essential oils and Funaben T. This information was added in signatures in Table 2 and Figure 4(2). (lines 126-134; 160-164)
  5. Lines 216-221 and lines 226-230 contain repeats. It should be corrected - the repetition has been deleted.
  6. Tables 3 and 4 describe the statistical analysis of the results not presented in the manuscript. Authors should imagine the results and their statistical analysis. In other words, the authors should provide a description of the chemical composition of essential oils and relate it to the fungicidal and fungistatic activity of each oil. The authors discuss the chemical composition of essential oils in the Discussion section, but it seems to me that the data on the chemical composition of essential oils presented in the table in the Methods section will improve the understanding of this manuscript - in the Materials and Methods section, we added a Table 5: Chemical composition of the tested essential oils

  1. Section Discussion.
  2. Lines 255 - 258 should be removed. This is an instruction from the template - we corrected this
  3. The abbreviation for essential oils –EsO and EOs - is used in different ways. It should be corrected - we corrected this (EsO).

Reviewer 2 Report

Brief: The aim of the study was to determine the effectiveness of selected seven commercial 16 essential oils (EOs) (grapefruit, lemongrass, tea tree (TTO), thyme, verbena, cajeput, and Litsea cubeba) on fungal isolates (Fusarium ) from  Central European obtained from 18 infected wheat kernels, and to evaluate the oils as potential natural fungicides. The study was conducted in 2 stages

Abstract: Some assertion is confusing in Abstract, can be simplified for better understanding of readers (line 20-25 & 32-33).

Keywords: Provide more appropriate to the field of research

Introduction: Introduction is written concisely covering latest literature available with most recent citations in the field.

  • Authors need to be specific in wheat diseases (rusts/powdery mildew/flag smut) they are referring (Line 54)
  • Insert “essential oils” (line 83).

Results: They are clear, presented in Table and figure formats.

Table 1: Authors need to maintain uniformity in giving nomenclature for fungal isolates across the text. It has been noted difference in Abstract and the fungal isolates names in the manuscript (refer Lines 404-407).

Typo error: cajeput or cayeput? check for elsewhere in the text (line 113)

As per Table 2: Control is "0" for isolate FC2D. cross check for typo errors.

Discussion: Authors need to interlink their findings with others and available literature. Section 3.1 need to be rewritten with proper linking to current findings in the research.

delete this statement (line 255-257)

Materials and Methods: The methods given in this paper are appropriate, clear with possibility of replication. They are well designed and executed. Also, advanced techniques are used to test the hypothesis.

  • Materials and methods: Lines 365-372 can be deleted. Copy and paste from authors guidelines?
  • Line 376: Reference must be cited if the fungal species are identified taxonomically (line 376) at this point.
  • amend the sentence -“ In the study were used commercial EsO” (Line 376)
  • support with citation (line 393)
  • clarify the statement as it is confusing(line 397). Whether the fusarium sp. is from Poland population? or German population?
  • Line 409: replace “ transplanted” with "subcultured". Authors are recommended to use the right scientific words and proper English language editing services while resubmission.
  • Line 424: Authors need to disclose whether experiments were carried out in triplicates? for clarity and repetition of experiments.

Conclusion: conclusion should highlight the current findings shedding light on future research. Authors need to emphasize more on this and improve in this reporting.

Author Response

We would like to thank the Reviewer for careful reading of this manuscript and for the comments and suggestions, which help to improve our manuscript. We have tried to do our best to respond to any comments and improve manuscript. As indicated below, we have made changes accordingly to all the general and specific comments. As suggested, the manuscript was expanded and the description of the techniques was refined. We added tables and the terms clarified. Changes and additions to the manuscript (marked in red).

Abstract: Some assertion is confusing in Abstract, can be simplified for better understanding of readers (line 20-25 & 32-33) - we corrected this (lines 16-39)

Keywords: Provide more appropriate to the field of research - we corrected this (lines 40-41)

Introduction: Introduction is written concisely covering latest literature available with most recent citations in the field. - thank you very much

Authors need to be specific in wheat diseases (rusts/powdery mildew/flag smut) they are referring (Line 54) - we corrected this – “In central Europe, the most dominant Fusarium ear blight causing species are F. graminearum, F.poae, F. avenaceum, F. culmorum, F. langsethiae and F. cerealis [5 – 7]”. (lines 57-59)

Insert “essential oils” (line 83) - we corrected this (lines 87)

Results: They are clear, presented in Table and figure formats - thank you very much

Table 1: Authors need to maintain uniformity in giving nomenclature for fungal isolates across the text. It has been noted difference in Abstract and the fungal isolates names in the manuscript (refer Lines 404-407) - we corrected this (Table 1 – line 112; lines 425-426; 454-458)

Typo error: cajeput or cayeput? check for elsewhere in the text (line 113) - we corrected this. It should be “cajeput”

As per Table 2: Control is "0" for isolate FC2D. cross check for typo errors. - In Table 2, checks the values of descriptive statistics between the rows corresponding to Funaben T and control have been converted. Of course, this mistake was corrected in the manuscript. (lines 127- 134)

Discussion: Authors need to interlink their findings with others and available literature. Section 3.1 need to be rewritten with proper linking to current findings in the research - we corrected this (lines 269-274; 318 -324)

delete this statement (line 255-257) - this part has been deleted.

Materials and Methods: The methods given in this paper are appropriate, clear with possibility of replication. They are well designed and executed. Also, advanced techniques are used to test the hypothesis. - thank you very much

Materials and methods: Lines 365-372 can be deleted. Copy and paste from authors guidelines? - this part has been deleted.

Line 376: Reference must be cited if the fungal species are identified taxonomically (line 376) at this point – we added reference (lines 424-427):

“At the first stage, the fungistatic activity of EsO was evaluated against four species of Fusarium from the Polish population (F. avenaceum FAPL, F. culmorum FCPL, F. graminearum FGPL and F. oxysporum FOPL) isolated from infected wheat kernels in south-western Poland in 2012–2014 [1].

- amend the sentence -“ In the study were used commercial EsO” (Line 376) - in the Material and Methods section, there is a Table 5 with the chemical compositions of essential oils used in both stages of the work.

support with citation (line 393)-clarify the statement as it is confusing(line 397). Whether the fusarium sp. is from Poland population? or German population? - in the Materials and Methods section, information was given (lines 431-436)

“At the second stage, on the basis of the mycelium growth rate index, the effectiveness of the EsO in limiting the rise of Fusarium isolates from the German population (F. culmorum FC1D, F. culmorum FC2D, F. graminearum FG1D, F. graminearum FG2D and F. poae FP0D) was assessed. The isolates were separated in 2012–2014 from infected wheat kernels obtained from Leibniz Zentrum für Agrarlandschaftsforschung e.V., Institut für Landschaftsbiogeochemie (ZALF, Müncheberg, Germany) collection”.

We added (lines 424-427): “At the first stage, the fungistatic activity of EsO was evaluated against four species of Fusarium from the Polish population (F. avenaceum FAPL, F. culmorum FCPL, F. graminearum FGPL and F. oxysporum FOPL) isolated from infected wheat kernels in south-western Poland in 2012–2014 [1].”

Line 409: replace “ transplanted” with "subcultured". Authors are recommended to use the right scientific words and proper English language editing services while resubmission - we corrected this (line 458)

Line 424: Authors need to disclose whether experiments were carried out in triplicates? for clarity and repetition of experiments. - in the Materials and Methods section, information was given “The tests were performed in four repetitions (n = 4), taking as a repetition one Petri dish from the inoculum in the form of a disc overgrown with pathogen mycelium” (lines 399-400).

Conclusion: conclusion should highlight the current findings shedding light on future research. Authors need to emphasize more on this and improve in this reporting - we corrected this (lines 489-494; 502-509)
